# Sex-specific expression profiles of ecdysteroid biosynthesis and ecdysone response genes in extreme sexual dimorphism of the mealybug *Planococcus kraunhiae* (Kuwana)

Miyuki Muramatsu[1], Tomohiro Tsuji[1], Sayumi Tanaka[1¤a], Takahiro Shiotsuki[2], Akiya Jouraku[3], Ken Miura[1], Isabelle Mifom Vea[1¤b], Chieka Minakuchi[1]*

1 Graduate School of Bio-Agricultural Sciences, Nagoya University, Nagoya, Japan, 2 Faculty of Life and Environmental Science, Shimane University, Matsue, Japan, 3 Institute of Agrobiological Sciences, National Agriculture and Food Research Organization, Tsukuba, Japan

¤a Current address: Kyushu Okinawa Agricultural Research Center, National Agriculture and Food Research Organization, Koshi, Japan
¤b Current address: Department of Biological Sciences, University of Illinois at Chicago, Chicago, IL, United States of America
* c_mina@agr.nagoya-u.ac.jp

**Data Availability Statement:** Obtained nucleotide sequences are available from the DDBJ/EMBL-

## Abstract

Insect molting hormone (ecdysteroids) and juvenile hormone regulate molting and metamorphic events in a variety of insect species. Mealybugs undergo sexually dimorphic metamorphosis: males develop into winged adults through non-feeding, pupa-like stages called prepupa and pupa, while females emerge as neotenic wingless adults. We previously demonstrated, in the Japanese mealybug *Planococcus kraunhiae* (Kuwana), that the juvenile hormone titer is higher in males than in females at the end of the juvenile stage, which suggests that juvenile hormone may regulate male-specific adult morphogenesis. Here, we examined the involvement of ecdysteroids in sexually dimorphic metamorphosis. To estimate ecdysteroid titers, quantitative RT-PCR analyses of four *Halloween* genes encoding for cytochrome P450 monooxygenases in ecdysteroid biosynthesis, i.e., *spook*, *disembodied*, *shadow* and *shade*, were performed. Overall, their expression levels peaked before each nymphal molt. Transcript levels of *spook*, *disembodied* and *shadow*, genes that catalyze the steps in ecdysteroid biosynthesis in the prothoracic gland, were higher in males from the middle of the second nymphal instar to adult emergence. In contrast, the expression of *shade*, which was reported to be involved in the conversion of ecdysone into 20-hydroxyecdysone in peripheral tissues, was similar between males and females. These results suggest that ecdysteroid biosynthesis in the prothoracic gland is more active in males than in females, although the final conversion into 20-hydroxyecdysone occurs at similar levels in both sexes. Moreover, expression profiles of ecdysone response genes, *ecdysone receptor* and *ecdysone-induced protein 75B*, were also analyzed. Based on these expression profiles, we propose that the changes in ecdysteroid titer differ between males and females, and that high ecdysteroid titer is essential for directing male adult development.

Bank/GenBank International Nucleotide Sequence Database (accession numbers: spo, LC508221; dib, LC508220; sad, LC508223–LC508225; shd, LC508222; EcR, LC508219; E75 isoforms, LC508214–LC508218).

**Funding:** IMV was funded by a fellowship from the Japan Society for the Promotion of Science (JSPS). This work was supported by JSPS KAKENHI Grant Numbers 15K07791 and 19H02969 to CM.

**Competing interests:** The authors have declared that no competing interests exist.

## Introduction

Arthropods (insects, arachnids, and crustaceans) and nematodes belong to Ecdysozoa, a super-phylum characterized by molting events–developing an exoskeleton that sheds during their development or lifetime. Among them, insects later acquired the ability to undergo metamorphosis which, combined to molting, this allowed insects to evolve highly diverse phenotypes and enabled them to conquer a wide range of habitats, contributing to their extraordinary diversity today [1]. Insect molting and metamorphosis are strictly regulated by two major hormones, juvenile hormone (JH) and ecdysteroids [2–4]. The titers of these hormones fluctuate throughout development, and work in concert to both determine the next state of developmental stage and molting event timing. As such, ecdysteroid pulses accompanied with high JH titer usually induce status quo molts, whereas ecdysteroid pulses with low JH titer induce metamorphic molts [5]. Thus, ecdysteroids are essential to trigger the successive molts throughout insect post-embryonic development [6].

The ecdysteroid biosynthetic pathway consists of a chain of enzymatic reactions. First, cholesterol from dietary sources is converted to 7-dehydrocholesterol (7dC) by a Rieske oxygenase Neverland [7, 8]. 7dC is then converted into 3β,14α-dihydroxy-5β-cholest-7-en-6-one (5β-ketodiol) through several steps. The detailed pathway from 7dC to 5β-ketodiol has not been clarified yet, hence named the "Black Box" [9–12]. To date, Spook (Spo, CYP307A1), Spookier (Spok, CYP307A2), CYP6T3, and Non-molting glossy/Shroud (Nm-g/Sro) were shown to be involved in the "Black Box" reactions [13–17]. Among the ecdysteroid biosynthetic genes, *sro* encodes a short-chain dehydrogenase/reductase, whereas *spo*, *spok*, and *cyp6T3* encode cytochrome P450 monooxygenases. *spo* and *spok* are paralogous gene copies, and have different functions in the fruit fly *Drosophila melanogaster* [13, 16]. Another paralogous gene, *spookiest* (*spot*, *Cyp307b1*), was identified in Coleoptera, Hymenoptera, and Diptera species other than Drosophilidae [16, 18]. The reaction from 5β-ketodiol to 2,22-dideoxyecdysone (5β-ketotriol) is catalyzed by Phantom (Phm, CYP306A1) [19, 20]. 5β-Ketotriol is converted into 2-deoxyecdysone by Disembodied (Dib, CYP302A1) [21, 22], and finally Shadow (Sad, CYP315A1) converts it into ecdysone [22]. Ecdysone is released from the prothoracic gland into the hemolymph and metabolized to 20-hydroxyecdysone (20E) by the ecdysone 20-monooxygenase Shade (Shd, CYP314A1) in peripheral tissues such as fat body, midgut, and Malpighian tubules [23].

The function of these ecdysteroidogenic enzymes was first studied in *D. melanogaster*: loss-of-function mutations of *spo*, *sro*, *phm*, *dib*, *sad*, or *shd* resulted in embryonic lethality, and a phenotype with naked and polished cuticles was observed [21]. Thus, these genes were named "*Halloween* genes" [24, 25]. Expression analysis of some of the *Halloween* genes in the silkworm *Bombyx mori* revealed that their developmental expression profiles were positively correlated with hemolymph ecdysteroid titer [15, 19, 26].

Mealybugs (Hemiptera: Pseudococcidae) are characterized by remarkable sexual dimorphism, as a result of unusual diverging post-embryonic development. In Japanese mealybug, *Planococcus kraunhiae* (Kuwana), males and females are phenotypically undistinguishable until the middle of the second nymphal instar (N2). After this stage, males metamorphose through non-feeding quiescent stages called prepupa (Pre) and pupa (Pu) into winged adults, whereas N2 females develop through successive molting events to the adult stage, retaining the wingless, juvenile features of nymphal instars. Hormonal regulation has been reported to be involved in sex-specific morphogenesis in some insects. For example, horn length in the dung beetle *Onthophagus taurus* is regulated by JH [27, 28], while the sex-specific mandible growth in the stag beetle *Cyclommatus metallifer* is affected by JH [29]. Moreover, sex-specific wing patterns in the butterfly *Bicyclus anynana* are controlled by diverging 20E titers [30]. Finally,

our previous study suggests that the JH titer in *P. kraunhiae* is lower in females than in males, so JH is likely to be involved in establishing sexual dimorphism in mealybugs [31]. We further showed that the adult specifying transcription factor *E93*, which is involved in in hormonal signaling pathways, is only expressed at the end of male adult development [32]. Since their titers have not been measured in *P. kraunhiae*, the involvement of ecdysteroids in sexual dimorphism remains unknown. In order to better understand the role of ecdysteroids in mealybug sex-specific post-embryonic development, measuring their titer is a critical step.

In this study, we examined the ecdysteroid titers in *P. kraunhiae* life cycle, with a focus on post-embryonic development. We initially attempted to measure the direct titers of ecdysteroids in pooled nymphs (approximately 200 individuals; ca. 10 mg in total) using liquid chromatography-tandem mass spectrometry (LC-MS/MS). Detection of ecdysteroids was unsuccessful probably because of their small body size (Muramatsu *et al.*, unpublished). We therefore estimated ecdysteroid titers by analyzing the expression profiles of the *Halloween* genes using quantitative RT-PCR. To further validate the estimated ecdysteroid titers, the expression profiles of ecdysone response genes, *ecdysone receptor* (*EcR*), and *ecdysone-induced protein 75B* (*E75*), were also measured. Our results suggest that ecdysteroid titer fluctuates in a sex-specific manner.

## Materials and methods

### Insect rearing conditions

The *P. kraunhiae* mealybugs were reared at 23°C (16L8D) on sprouted broad beans (Kokusai Pet Food, Kobe, Japan) as described in a previous study [31]. In these conditions, from egg oviposition to adult emergence, development times were approximately as follows: 9–11 days for the embryonic stage (E) after oviposition, 11 days for the first-instar nymphs (N1), 4 days for the phenotypically undifferentiated second-instar nymphs (N2), 3–4 days for the differentiated female second-instar nymphs (N2♀), 5 days for differentiated second-instar nymphs (N2♂), 9 days for the female third-instar nymphs (N3), 4 days for the male prepupae (Pre), and 5–6 days for the male pupae (Pu).

### Sex ratio estimation and collection strategy of staged individuals

Males and females of *P. kraunhiae* from E to N2D3 are not distinguishable from their external morphology. However, because the sex ratio of the eggs that a female lays depends on oviposition time, sex-biased eggs can be collected at different oviposition days, as previously reported [31, 33]. Samples from E to N2D3 for quantitative RT-PCR were therefore collected using the sex-biased strategy as follows: eggs laid on day 1 of oviposition were collected as male-biased samples, and eggs laid on day 5 of oviposition were collected as female-biased samples. In order to obtain staged nymphs, mated adult females were separated in glass dishes containing a sprouted broad bean, and the eggs were collected every 24 hours and monitored for development in separate glass dishes until they attained the desired stage. Staged pooled individuals were then homogenized in TRIzol reagent (Thermo Fisher Scientific Inc., MA, USA) for RNA extraction. In order to confirm the sex ratio, some individuals were left to develop until N3 or Pre stages for observation.

### cDNA cloning of *Halloween* genes and ecdysone response genes in *P. kraunhiae*

To identify homologous sequences of *spo*, *phm*, *dib*, *sad* and *shd* in *P. kraunhiae*, tblastn searches were performed using the unpublished RNA-seq database of *P. kraunhiae* from our

laboratory (Vea *et al.*, unpublished) with the amino acid sequences of other insects, as listed in S1 Table. Similarly, tblastn searches were performed in the RNA-seq database (DDBJ/EMBL-Bank/GenBank accession number DRA004114) [34] using *EcR* and *E75* sequences from other insects.

Total RNA was extracted from pooled individuals of different stages and sexes using TRIzol reagent as reported previously [31]. Oligo-dT-primed reverse transcription was performed with PrimeScript II 1st strand cDNA synthesis kit (Takara Bio Inc., Shiga, Japan). Primers for RT-PCR were designed based on putative nucleotide sequences identified in RNA-seq databases. PCR products were purified and subcloned into pGEM-T Easy Vector (Promega Corp., WI, USA) and sequenced. To obtain the complete nucleotide sequences of *E75* variants at the 5' end, 5' RACE PCR was performed with a SMARTer RACE cDNA Amplification Kit (Takara Bio) as previously reported [31, 32].

Primer sequences are listed in S2 Table. Obtained nucleotide sequences were deposited in the DDBJ/EMBL-Bank/GenBank International Nucleotide Sequence Database with the following accession numbers: *spo*, LC508221; *dib*, LC508220; *sad*, LC508223–LC508225; *shd*, LC508222; *EcR*, LC508219; *E75* isoforms, LC508214–LC508218.

To confirm the homology of the candidate *Halloween* genes (cytochrome P450 gene family), we aligned the translated amino acid sequences of all *P. kraunhiae Halloween* genes with known sequences of other insects, using MAFFT v.7 via the online service [35] and using the L-INS-i method [36]. The phylogeny was then inferred using the Bayesian method, and were carried out with MrBayes v3.2.6 [37] using the mixed amino acid model, through the Cipres Science Gateway Portal [38]. Four independent runs were carried out for 1 million generations each, and trees were sampled every 100 generations. After the analysis, the phylogeny was estimated based on the majority consensus of sampled trees, after removing the first 25% trees (burn-in). The nexus file containing the sequence alignment and analysis script can be found in S1 File.

## Gene expression

Absolute quantitative RT-PCR was performed as described previously [31]. Briefly, samples were collected every 24 h after oviposition up to adult emergence. From E to N2D3 were collected using a sex-biased strategy. Total RNA was extracted from pooled individuals as described above. These RNA samples were reverse transcribed using a Prime Script RT reagent Kit with gDNA Eraser (Takara Bio). Quantitative RT-PCR was carried out in a 14 μl reaction volume containing SYBR Ex Taq (Takara Bio), 0.2 μM of each primer (see S2 Table) and 1 μl of template cDNA or standard plasmids. PCR conditions were 95˚C for 30 s, followed by 40–45 cycles at 95˚C for 5 s and 60˚C for 30 s. After thermal cycling, the absence of unwanted byproducts was confirmed using melting curve analysis. Serial dilutions of a plasmid containing a part of the ORF of each gene were used as standards. Transcript levels of the target genes were normalized to that of *ribosomal protein L32* (*rpL32*) levels in the same samples.

## Results

### *P. kraunhiae Halloween* genes cloning and expression profiles

The search for *Halloween* gene homologs in the *P. kraunhiae* transcriptome retrieved different candidate transcripts. Using designed primers, we performed RT-PCR on cDNA synthesized from total RNA extracted from pooled individuals of different stages and obtained the partial sequences of the following genes: a 953-bp long *Pkspo* transcript, a 1286-bp *Pkdib* transcript, and a 1640-bp *Pkshd* transcript. We retrieved three variants of *Pksad* (here designated as variants A, B and C): variant-A (1535 bp), variant-B (with a 20-bp deletion compared with

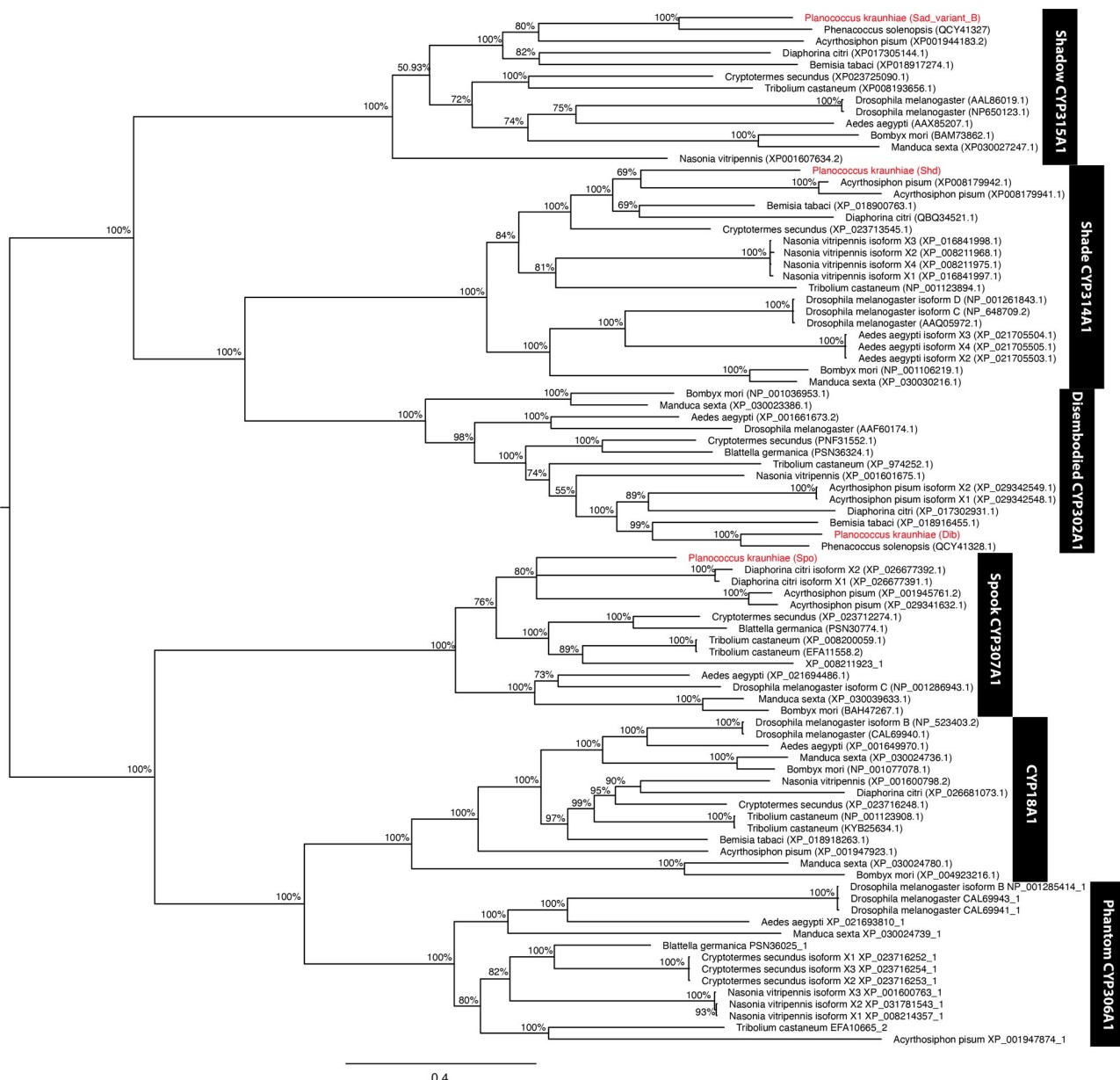

**Fig 1. Cytochrome P450 gene family phylogeny of representatives from different insect orders.** The phylogenetic reconstruction was inferred using MrBayes based on the majority consensus obtained from 4 runs at 1 million generations, with trees sampled every 100 generations, using the mixed amino acid model. Posterior probability values are indicated at each node. Genes from *Planococcus kraunhiae* are highlighted in red in the phylogeny.

variant-A), and variant-C (which had an additional 146-bp deletion). *Pksad* transcripts are likely a result of alternative splicing and a blastp search of the predicted protein sequences retrieved variant-B with the highest homology to other insect functional sad proteins (data not shown).

To further confirm the homology of the identified *Halloween* genes, the predicted amino acid sequences were aligned with *Halloween* genes from different insect species and a phylogenetic tree was inferred using the Bayesian method. Our phylogenetic tree retrieved the different *Halloween* genes of *P. kraunhiae* into their respective groups (Fig 1). Moreover, the amino

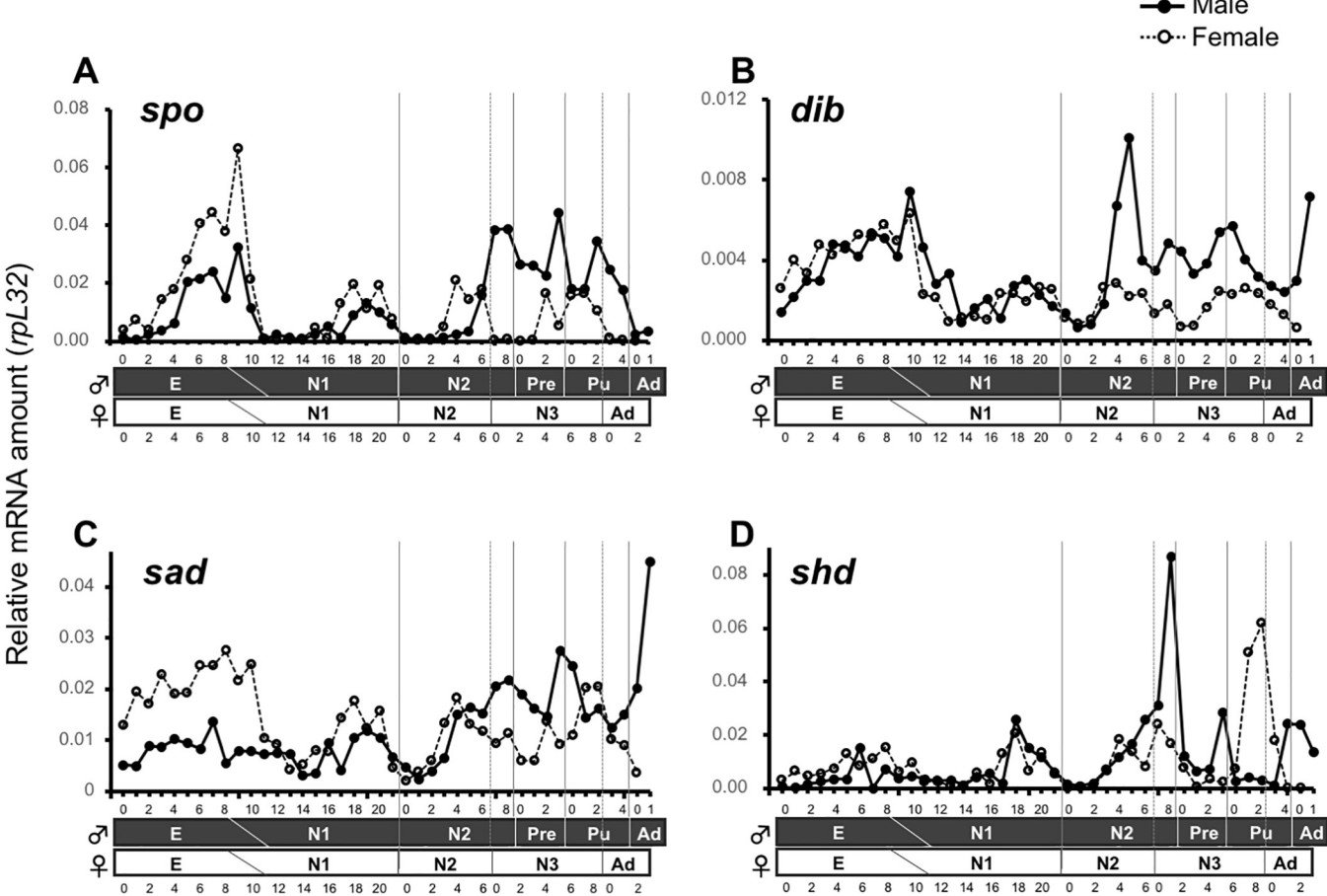

**Fig 2.** Developmental expression profiles of *Planococcus kraunhiae spook* (A), *disembodied* (B), *shadow* (C), and *shade* (D). Transcript levels were determined using absolute quantitative RT-PCR, and the values were normalized to those of *PkrpL32*. RNA was isolated from pooled individuals. E: Egg, N1: first-instar nymph, N2: second-instar nymph, N3: female third-instar nymph, Pre: male prepupa, Pu: male pupa, Ad: adult. Numbers on x-axis indicate the ages in days within each developmental stage. Solid circles represent males, while open circles represent females.

acid sequences were aligned for each *Halloween* gene with the sequences of *D. melanogaster*, *Tribolium castaneum* and *B. mori* and showed that these genes are highly conserved (S1 Fig). For instance, in PkSpo, the alignment highlighted the conserved signature sequences of a cytochrome P450 protein, such as "PERF" domain (PxxFxPxRF) and heme-binding domain (PFxxGxRxCxG) (S1 Fig). Among the *Halloween* genes involved in ecdysteroid biosynthesis, we were not able to identify a candidate sequence for *phm* in *P. kraunhiae*.

To compare ecdysteroid titers between male and female developments, we indirectly measured the titers by quantifying transcript levels of *Pkspo*, *Pkdib*, *Pksad*, and *Pkshd* every 24 h from oviposition to adult emergence in both sexes (Fig 2).

*Pkspo*, *Pkdib* and *Pksad* were highly expressed during embryonic development after oviposition and sex-specific expression was found in *Pkspo* and *Pksad*, where females had higher levels of transcripts (Fig 2). The N1 stage did not show notable expression differences between *Halloween* genes or sexes, except for very low expression of *Pkspo* and *Pkshd* in the first few days after hatching. Most of the sex-specific expression differences were found starting at the end of N2. Generally, all *Halloween* genes had higher expression levels during male development, with peaks preceding molting events. Although in a lesser extent, the same pattern of expression timed to molting events also occurred in females, with notably two peaks of *Pkspo*

and *Pkshd* at the end of N2 and N3. Finally, *Pkshd* expression profile presented a distinct expression pattern compared to the other *Halloween* genes: expression levels remained low from E to the middle of N1, increasing during the last four days of N1, before molting to N2, in both sexes. Interestingly, *Pkshd* mRNA highest peaks coincided with the onset of metamorphosis for males (before prepupal stage), and the onset of adult molting event, at the end of N3 for females (Fig 2D).

## Expression of ecdysone response genes in *P. kraunhiae*

To further assess how ecdysone is involved in the establishment of extreme sexual dimorphism, we cloned and measured the expression of *ecdysone receptor* (*EcR*) and one of the early response genes in the ecdysone signaling pathway *E75*. Using RT-PCR, a 1565-bp fragment for *PkEcR* was amplified and sequenced, which revealed that the amino acid sequence of PkEcR is most similar to EcR-A isoforms in other insect species (S2 Fig). Regarding *E75*, cDNA sequences of five variants, generated from different transcription initiation sites and alternative splicing, were obtained by RT-PCR and 5' RACE PCR (Fig 3). These variants were named *E75A*, *E75B*, *E75C*, *E75D*, and *E75E*, based on the homology with other insect counterparts. An alignment of the E75A amino acid sequences (S3 Fig) showed that these sequences were conserved especially within the putative DNA-binding and ligand-binding domains.

 *PkEcR* expression coincides with hatching and molting events throughout both male and female development (Fig 4A). Between the embryonic stage and the end of N2, *PkEcR* expression was progressive and happened as small peaks at hatching and N1–N2 transition, with

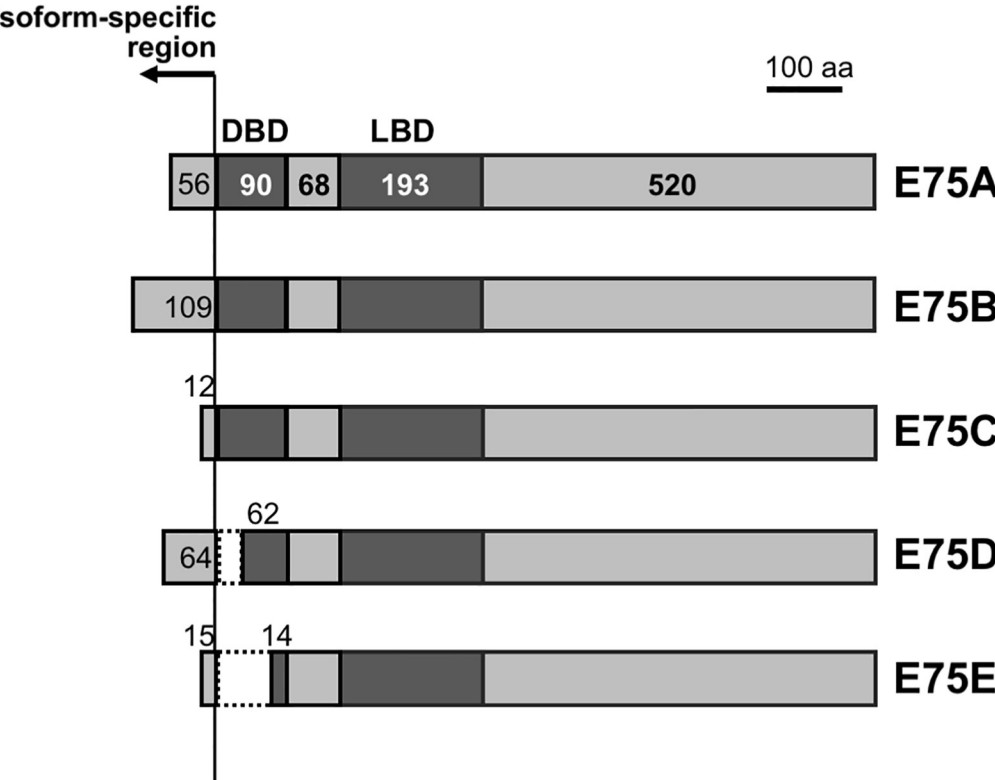

**Fig 3. Structure of *Planococcus kraunhiae* E75 isoforms.** Open reading frames are shown as boxes. The putative DNA-binding domain and ligand binding domain are shown in dark grey. Numbers in boxes indicate amino acids for each domain. E75D lacks 28 amino acids in putative DBD, whereas E75E lacks 74 amino acids.

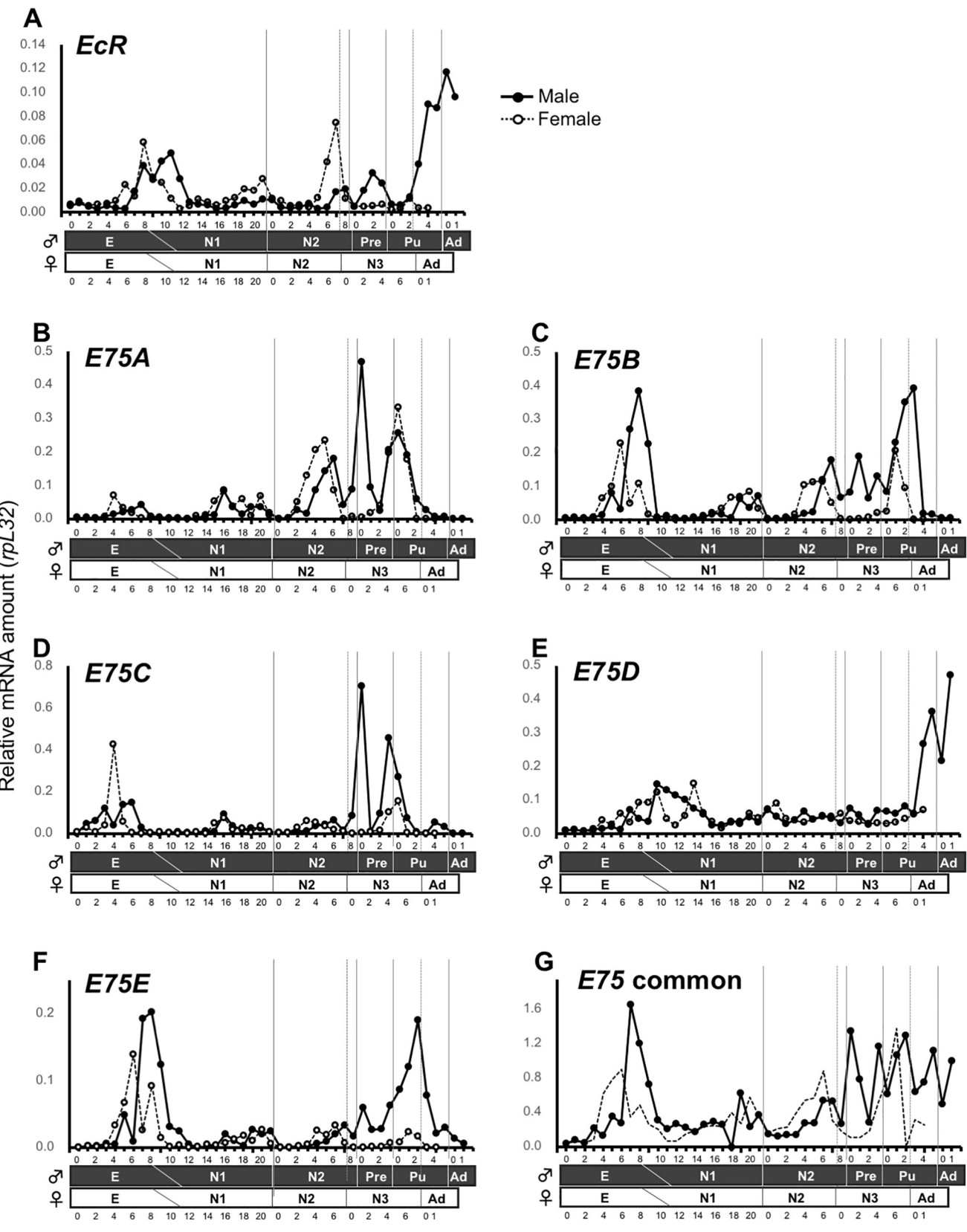

**Fig 4. Developmental expression profiles of *Planococcus kraunhiae E75* and *EcR*.** Transcript levels of *EcR* (panel A), *E75A* (panel B), *E75B* (panel C), *E75C* (panel D), *E75D* (panel E), *E75E* (panel F), and *E75* common region (panel G) were determined by quantitative RT-PCR, and the values were normalized to those of *PkrpL32*. RNA was isolated from pooled individuals. E: Egg, N1: first-instar nymph, N2: second-instar nymph, N3: female third-instar nymph, Pre: male prepupa, Pu: male pupa, Ad: adult. Numbers on the x-axis indicate the ages in days within each developmental stage. Solid circles represent males, while open circles represent females.

female transcripts being slightly higher. From N2, *PkEcR* levels in males increased progressively at each molting event, reaching the highest expression at the Pu–Ad transition. In females, however, the highest peak of expression occurred at the end of female N2, but remained very low during the N3–Ad transition.

The expression profiles of *PkE75* isoforms markedly differed among each other (Fig 4B–4F). For instance, *PkE75A* and *PkE75C* showed a distinct male-specific peak of expression at the N2–Pre transition (Fig 4B and 4D), although there was a transient peak in females at Day4 of the embryonic stage. Alternatively, the peak of *PkE75B* coincided to molting events in both males and females, although males peaks were generally higher (Fig 4C). *PkE75D* was highly expressed at the male Pu–Ad transition, while there was no obvious peak in other developmental stages as well as during female development (Fig 4E). Finally, the transcript levels of *PkE75E* increased during the latter half of each instar, and was prominent at the end of E, as well as during male Pre and Pu stages (Fig 4F). The expression profile with primers for *PkE75* common region reflected those of the five isoforms: it was high in the latter half of each instar, and prominent in male Pre to Ad (Fig 4G).

Taken together, the transcript levels of *PkE75* isoforms, especially *PkE75C* and *PkE75E*, were higher at the onset of male metamorphosis (Pre and Pu stages) compared to females, while *PkE75D* was only high at the male Pu–A transition.

## Discussion

The role of ecdysteroids in insect metamorphosis is already extensively investigated in selected insect models that undergo complete metamorphosis, such as *D. melanogaster* and *B. mori*. However, little is known of ecdysteroid involvement in hemimetabolous insects, and more specifically how it establishes sex-specific metamorphosis in mealybugs. The goal of this study was to first provide evidence of how ecdysteroid titer is linked to sexually dimorphic development in the mealybug *P. kraunhiae*. Because our attempts to directly measure ecdysteroids using LC-MS/MS methods were unsuccessful, we estimated indirectly ecdysteroid titers using the expression profiles of *Halloween* genes, as well as those of *EcR* and *E75*.

Expression of ecdysteroid biosynthetic genes and ecdysone response genes were previously assessed in another mealybug, *Phenacoccus solenopsis*, by a transcriptome analysis but was limited to a few developmental stages [39]. Here we present, for the first time, a detailed developmental expression pattern of ecdysteroid biosynthesis and ecdysone response genes, highlighting the major differences in gene expression between male and female development in the mealybug *Planococcus kraunhiae*. We first examined the developmental expression profiles of *spo*, *dib*, *sad*, and *shd*, *Halloween* genes that are highly conserved in arthropod groups [10]. In the silkworm *B. mori*, expression profiles of *Halloween* genes are positively correlated with the hemolymph ecdysteroid titer throughout development [15, 19, 26]. This suggested that the transcript levels of *Halloween* genes can be used as a good indicator of the ecdysteroid titer. We found that the transcript levels of *Pkspo*, *Pkdib*, *Pksad*, and *Pkshd* usually start increasing during the second half of N1 and N2 stages (Fig 2). This indicates that ecdysteroid biosynthesis in the prothoracic gland becomes active before each nymphal molt in *P. kraunhiae*. Importantly, from the middle of the N2 stage, when sexual dimorphism becomes visible, the expression profiles of *Pkspo*, *Pkdib*, *Pksad*, and *Pkshd* start differing between males and

females. In particular, the transcript levels of *Pkspo*, *Pkdib*, and *Pksad* remain higher in males compared with females from mid-N2 to the adult. We conclude that ecdysteroid biosynthesis in the prothoracic gland is more active in males than in females.

It is worth mentioning that the developmental expression profile of *Pkshd* was somewhat different from those of the other three *Halloween* genes tested: only *Pkshd* expression peaks in females at N3–Ad molting, which was not observed in *Pkspo*, *Pkdib*, or *Pksad* (Fig 2). In other insect species such as *D. melanogaster* and *Schistocerca gregaria*, Spo, Dib and Sad are generally located in the prothoracic gland, whereas Shd converts ecdysone into 20E by hydroxylation in peripheral tissues such as the fat body, midgut, and Malpighian tubules [23, 40]. Therefore, we propose that the conversion of ecdysone to 20E occurs before adult metamorphosis in both sexes, i.e., at the end of N2 for males and at the end of N3 for females. Nevertheless, since the transcript levels of *Pkspo*, *Pkdib*, and *Pksad* from mid-N2 to the adult were higher in males compared with those of females, the total amount of ecdysteroids in males might be higher than in females. In contrast, a high expression of *Pkshd* in females at N3–adult molting would be necessary for transient peak of 20E, which might be essential to induce female adult differentiation such as ovarian development. In addition, our results suggested that the transcription of *Pkspo*, *Pkdib*, and *Pksad*, the ecdysteroidogenic enzymes involved in the biosynthetic pathway from 7dC to ecdysone, is regulated in a similar manner, whereas the transcriptional regulation of *Pkshd* is distinct from the others. This might be because the tissue localization is different between *shd* and others.

We also examined the expression profiles of *E75* and *EcR*, both of which are known as ecdysone response genes in several insect species and are well understood in *D. melanogaster* [41] and *B. mori* [42, 43]. The transcript levels of the *E75* common region and *EcR* were high in male Pu stage (Fig 4A and 4G), suggesting that ecdysteroid titer is high during male adult development. Using 5' RACE PCR, we identified five isoforms of *E75* in *P. kraunhiae* as shown in Fig 3. Although the developmental expression profiles of these *E75* isoforms were generally similar, peaks of expression shifted slightly among isoforms (Fig 4). Similar observations have been reported in other species such as *Manduca sexta* [44] and *B. germanica* [45]. These isoforms must have distinct roles in insect development, and the transcriptional regulation mechanism differs among isoforms. For instance, involvement of JH in regulating *E75* transcription has already been reported, where JH suppresses 20E-induced transcription of *E75C* in adult development of *M. sexta* [44]. We reported previously that JH levels were higher in males during metamorphosis in *P. kraunhiae* [31]. Therefore, we suggest that the sex-specific expression of *PkE75* isoforms could be regulated by both ecdysteroids and JH.

Based on our qRT-PCR results, we estimated the ecdysteroid titers throughout post-embryonic development of *P. kraunhiae*. Among the genes that we examined in this study, we selected *Halloween* genes and *EcR* as indicators to measure ecdysteroid titer indirectly. As stated above, the transcription of *E75* isoforms seems to be regulated by both ecdysteroids and JH in isoform-specific manners, which makes it difficult to estimate ecdysteroid titer from the expression profiles of *E75* isoforms alone. As shown in Fig 5, in male development, there are peaks of ecdysteroids, which are likely to induce metamorphic molts to Pre, Pu and Ad. We propose that high ecdysteroid titer in males is essential to activate transcription factors such as *br* and *E93*. *br* is a pupal specifier in holometabolous insects, whereas it is involved in progressive wing formation in hemimetabolous species [46–51]. *E93* is a transcription factor that induces adult morphogenesis in both hemimetabolous and holometabolous species [52, 53]. It has been reported that the transcription of both *br* and *E93* is regulated by ecdysteroids and JH [51–54]. In *P. kraunhiae*, *br* expression is higher in males than in females, while *E93* is exclusively expressed during male adult metamorphosis [31, 32]. Higher ecdysteroids during male adult development would have a significant role in promoting adult morphogenesis through

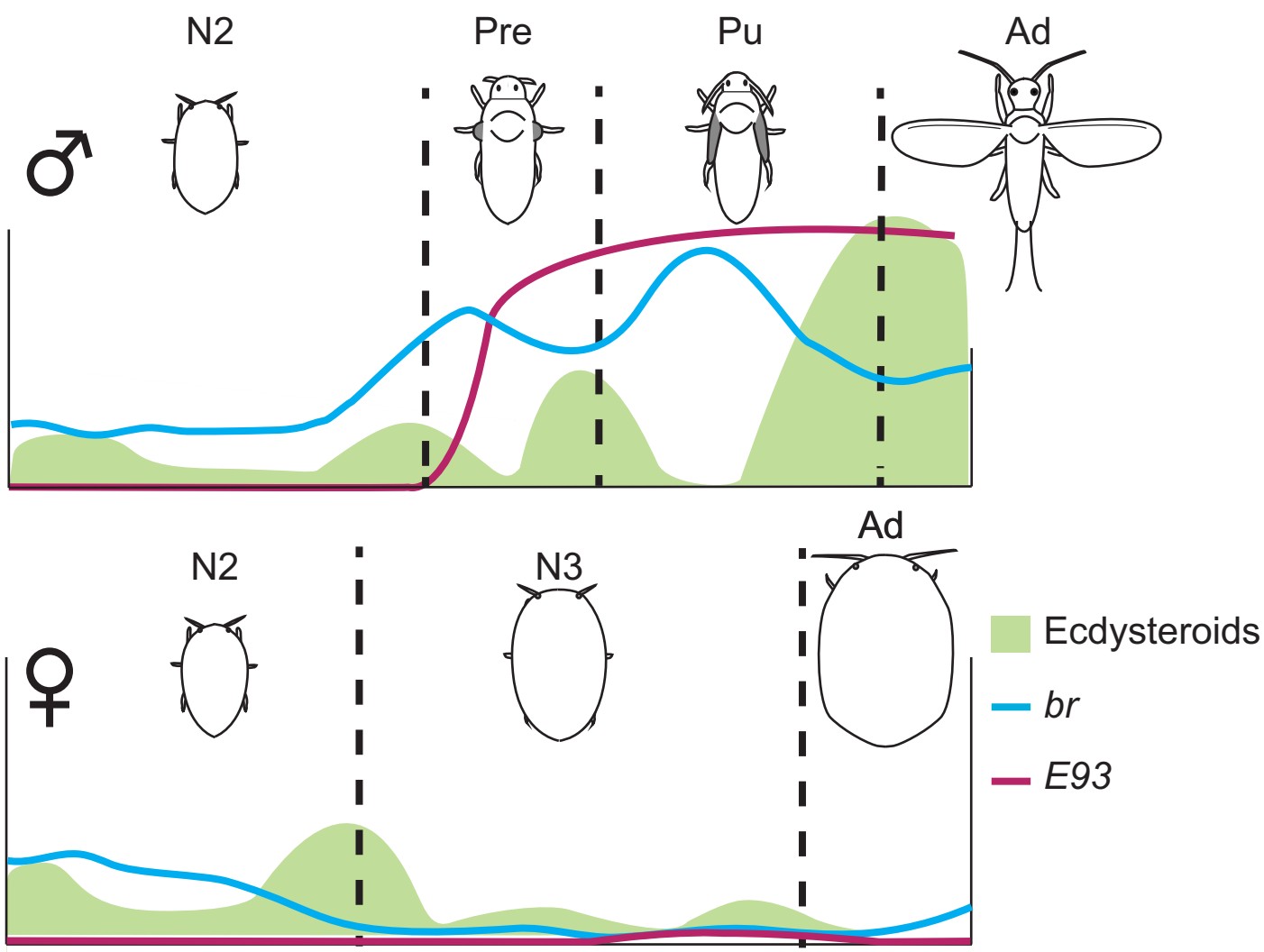

**Fig 5. Diagram of estimated ecdysteroid titer in *Planococcus kraunhiae* based on our study.** Ecdysteroid titer is shown in green. The expression profiles of *br* (blue line) and *E93* (magenta line) are based on our previous studies [31, 32]. N2: second-instar nymph, Pre: male prepupa, Pu: male pupa, N3: female third-instar nymph, Ad: adult.

*br* and *E93* (Fig 5, upper panel). In females, by contrast, ecdysteroid titer remains relatively low compared with males, although a transient peak is observed at N2-N3 transition (Fig 5, lower panel). The overall low ecdysteroid titer in females might account for their neotenic development and wingless adult stages.

The reason why our attempts to measure ecdysteroid titers using LC-MS/MS were not successful is not clear. One possibility is that due to their small body size, especially during the juvenile stages, it is not possible to collect hemolymph from the mealybugs, which might decrease the purity of extracted ecdysteroids for the analysis. Another possibility is the involvement of metabolism of ecdysteroids in the hemolymph of mealybugs: in the insect body, a part of the ecdysteroids isgenerally metabolized into polar metabolites such as esters and conjugates [55]. It is possible that most of the ecdysteroids in mealybugs are rapidly metabolized, which makes it difficult to identify ecdysteroids by LC-MS/MS. In order to extract enough amount of

ecdysteroids, it will be necessary to collect a higher number of individuals in which ecdysteroids biosynthesis is active: collecting individuals prior to ecdysis might help for this purpose.

In summary, our results suggest that the changes in ecdysteroid titer are diverse between females and males, and that higher ecdysteroids in males may play a significant role in promoting male-specific adult morphogenesis. Taken together with our previous studies [31, 32], we conclude that both ecdysteroids and JH play an essential role in establishing sexually dimorphic metamorphosis of mealybugs. Further studies such as promoter analysis of *br* and *E93* should provide insights into any crosstalk between ecdysteroids and JH.

## Supporting information

**S1 Fig. Alignment of protein sequences of spook, disembodied, shadow and shade.** Protein sequences of Spook (A), Disembodied (B), Shadow (C) and Shade (D) were aligned among *Planococcus kraunhiae* (Pk), *Bombyx mori* (Bm), *Drosophila melanogaster* (Dm) and *Tribolium castaneum* (Tc). Putative "PERF" domains as well as heme-binding domains were indicated with lines. Accession numbers follow gene names.
(PDF)

**S2 Fig. Alignment of EcR protein sequences.** BgEcR, *Blattella germanica* EcR (accession number, CAJ01677.1); TcEcR, *Tribolium castaneum* EcR-A (NP_001107650.1); NvEcR, *Nezara viridula* EcR-A (ADQ43370.1); PkEcR, *Planococcus kraunhiae* EcR (this study). Asterisks indicate fully-conserved amino acid residues, while colons and periods represent conservation with strong and weak similarity, respectively. The DNA binding domain (C region) and ligand binding domain (E region) are boxed. The putative junction between EcR-A and EcR-B isoforms is shown by an arrow.
(PDF)

**S3 Fig. Alignment of E75A protein sequences.** PkE75A, *Planococcus kraunhiae* E75 isoform A (this study); BgE75A, *Blattella germanica* E75A (accession number, CAJ87513.1); BmE75A, *Bombyx mori* E75A (NP_00106079.1). Asterisks indicate fully-conserved amino acid residues, while colons and periods represent conservation with strong and weak similarity, respectively. The DNA-binding domain and ligand-binding domain are boxed.
(PDF)

**S1 Table. Accession numbers of the query amino acid sequences of *Halloween* genes.**
(PDF)

**S2 Table. Primer sequences for RT-PCR, RACE PCR and qRT-PCR.**
(PDF)

**S1 File. Nexus file of aligned Cytochrome P450 genes of different insect species.** Nexus format file including aligned amino acid sequences of cytochrome P450 gene regions and command lines used for the MrBayes phylogenetic inference.
(NEX)

## Acknowledgments

We thank Dr. Hajime Ono (Kyoto University) for LC-MS/MS analyses. We also thank Dr. Jun Tabata for providing us the Japanese mealybug stocks.

## Author Contributions

**Conceptualization:** Isabelle Mifom Vea, Chieka Minakuchi.

**Data curation:** Miyuki Muramatsu, Tomohiro Tsuji, Takahiro Shiotsuki, Akiya Jouraku, Chieka Minakuchi.

**Formal analysis:** Isabelle Mifom Vea.

**Funding acquisition:** Chieka Minakuchi.

**Investigation:** Tomohiro Tsuji, Sayumi Tanaka, Takahiro Shiotsuki, Chieka Minakuchi.

**Supervision:** Chieka Minakuchi.

**Validation:** Miyuki Muramatsu, Sayumi Tanaka, Akiya Jouraku, Isabelle Mifom Vea, Chieka Minakuchi.

**Visualization:** Miyuki Muramatsu, Chieka Minakuchi.

**Writing – original draft:** Miyuki Muramatsu, Isabelle Mifom Vea, Chieka Minakuchi.

**Writing – review & editing:** Miyuki Muramatsu, Takahiro Shiotsuki, Akiya Jouraku, Ken Miura, Isabelle Mifom Vea, Chieka Minakuchi.

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
