## [Decision Letter · Decision Letter 0]

4 Feb 2020

PONE-D-20-01381

Sex-specific expression profiles of ecdysteroid biosynthesis and ecdysone response genes in extreme sexual dimorphism of the mealybug Planococcus kraunhiae (Kuwana)

PLOS ONE

Dear Dr. Minakuchi,

Thank you for submitting your manuscript to PLOS ONE. After careful consideration, we feel that it has merit but does not fully meet PLOS ONE’s publication criteria as it currently stands. Therefore, we invite you to submit a revised version of the manuscript that addresses the points raised during the review process.

Reviewer 1 suggests developing an analytical method to directly detect and quantify ecdysteroids from mealybug by increasing the number of individuals analyzed. As both reviewer 2 and 3 are fully accepting of the use of expression profiles as proxies for titer and reviewer 3 clearly highlights the challenges that conjugation of ecdysteroids present in quantification, I believe reviewer 1’s request is beyond the scope of the current manuscript.  Please address the remaining concerns of all three reviewers, including comment 2 from reviewer 3 and pay particular attention to indicating biological replicate data in your qPCR results. 

We would appreciate receiving your revised manuscript by Mar 20 2020 11:59PM. To enhance the reproducibility of your results, we recommend that if applicable you deposit your laboratory protocols in protocols.io, where a protocol can be assigned its own identifier (DOI) such that it can be cited independently in the future. For instructions see: http://journals.plos.org/plosone/s/submission-guidelines#loc-laboratory-protocols

We look forward to receiving your revised manuscript.

Kind regards,

Christopher N. Boddy, Ph.D.

Academic Editor

PLOS ONE

Journal Requirements:

2. In your Methods section, please provide additional details regarding the insects used in your study and ensure you have described the source. For more information regarding PLOS' policy on materials sharing and reporting, see https://journals.plos.org/plosone/s/materials-and-software-sharing#loc-sharing-materials.

Reviewers' comments:

Reviewer's Responses to Questions

**Comments to the Author**

1. Is the manuscript technically sound, and do the data support the conclusions?

Reviewer #1: Partly

Reviewer #2: Yes

Reviewer #3: Yes

2. Has the statistical analysis been performed appropriately and rigorously? 

Reviewer #1: No

Reviewer #2: N/A

Reviewer #3: Yes

3. Have the authors made all data underlying the findings in their manuscript fully available?

Reviewer #1: Yes

Reviewer #2: Yes

Reviewer #3: Yes

4. Is the manuscript presented in an intelligible fashion and written in standard English?

Reviewer #1: No

Reviewer #2: Yes

Reviewer #3: Yes

5. Review Comments to the Author

Reviewer #1: Manuscript #: PONE-D-20-01381

Title: "Sex-specific expression profiles of ecdysteroid biosynthesis and ecdysone response genes in extreme sexual dimorphism of the mealybug Planococcus kraunhiae (Kuwana)"

Authors: Miyuki Muramatsu et al.

Comments

Mealybugs are characterized by remarkable sexual dimorphic traits as a result of unusual diverging post-embryonic development. In some insect species, such sexual dimorphic traits were regulated by hormones such as JH and ecdysteroids. However, ecdysteroid titers have not been measured in the Japanese mealybug, Planococcus kraunhiae, and thus the involvement of ecdysteroids in sexual dimorphism in this species remains unknown.

In order to cultivate a better understanding for the role of ecdystderoids in sexual-dimorphic development in the mealybug, the authors first attempted to directly measure the ecdysteroid titers using LC-MS/MS. But they failed to detect ecdysteroids by their method. Therefore, the authors tried to estimate ecdysteroid titers by quantifying mRNA levels of the Halloween genes together with ecdysone response genes such as EcR and E75 by qRT-PCR. Finally, they propose that the changes in ecdysteroid titer are different between males and females, and that high ecdysteroid titer is essential for directing male adult development.

Overall, this is a nicely done, and several interesting observations are represented and discussed. But I think that there are several unignorable problems especially in their qRT-PCR data. Moreover, English used in this manuscript has a lot of problems. Resolution of the figures is so poor that I cannot precisely recognize characters described on them. Consequently, this manuscript is not suitable for publication in this current form.

Major requirements for revision

1. page 4, line 109. The authors said that they were not able to detect any ecdysteroid, probably due to the small size of the mealybug. If so, then the authors should increase the number of individuals subjected to the LC-MS/MS analysis.

2. page 9, lines 242-244 and Fig. S2. The authors identified an EcR ortholog of the mealybug, which showed highest similarity to EcR-A isoforms in other insect species. Were there any other EcR isoforms such as EcR-B1 and EcR-B2? If not, then the authors should show several lines of evidence that the mealybug examined in this study has only one isoform. And also, they should perform phylogenetic analysis as described in Fig. 1 to evaluate whether the PkEcR is indeed an ortholog of EcR in the mealybug.

3. Fig 2 and 3. How many trials did the authors perform qRT-PCR analyses? In general, researchers perform at least triplicate trials of qRT-PCRs and give SE to statistically assess the quantified data. The authors should provide such kind of information about statistical analysis on their qRT-PCR data.

4. page 10, lines 272-273. The authors said that expression levels of PkE75C and PkE75E were higher in males than females at the onset of male metamorphosis. But the expression level of PkE75C was specifically higher in females than males at day-4 embryo. The level was as same as that in male at pre-pupal stage. The authors should give some explanations about this female-specific increment of PkE75C expression level.

5. page 12, lines 343-345. The authors said that the sex-specific expression of PkE75 isoforms could be regulated by both ecdysteroids and JH. If so, then expression levels of PkE75 isoforms do not simply reflect ecdysteroids titers. This means that people cannot estimate ecdysteroid titers based on the expression levels of PkE75 isoforms. Therefore, the authors should directly quantify the ecdysteroid titers.

Minor requirements for revision

1. Overall. All text describing experimental results should be written in the past tense. For example, in the line 9, the author described that "PkEcR expression is progressive and happens as small peaks at hatching". But they should change this phrase to " PkEcR expression was progressive and happened as small peaks at hatching.

2. page 4, line 88. I think "a result of" should be better to change to "as a result of".

3. page 4, line 97. Add period immediately after "[29]".

4. page 4, line 101. I think "which is" should be better to insert prior to "involved in".

5. page 4, line 107. "directs" should be replaced with "direct".

6. page 6, line 166. "(" before [37] should be removed.

Reviewer #2: Sexual dimorphism on metamorphosis is known in meal bug. Generally, juvenile hormone (JH) and ecdysteroid play crucial roles on regulation of polypenism. Authors has already showed the JH titer of Planococus kraunhiae. In this manuscript, authors estimate the ecdysteroid profiles of male and female in the meal bug, P. kraunhiae. It is impossible to measure the ecdysteroid titer directly because the bug is too small to get enough volume of hemolymph. The ecdysteroid titer was estimated from the expression profiles of 4 ecdysterodgenic genes and 2 ecdysone responsible genes. According to data, they discussed ecdysteroid titer in both sex and speculates the role of ecdysteorid on sexual dimophism. The manuscript is written well and statement is clear.

In general, I do not see serious problems on the scientific aspects of the manuscript. Thus, I’d add only some comments.

1. In figure 5, ecdysteroid titer was estimated. The ecdysteroid level of female seems too low levels in N3 nymph. The expression levels of spo, dib and sad in N3 were similar to these in N2. What did you calculated the level is based on?

2. Line 97. Followed by [97], please add full stop.

3. Lin 100. What “the transcription factor” does indicate?

4. Line 107. I think it is mistype, please correct “the directi titers”.

5. Line 166. Please remove “(“.

6. Line 176. About what amount total RNA did you use for RT? Please mention.

7. In discussion section, from line 312 to 318 is very similar to line 319-330. Please rewrite these paragraphs.

8. In figure 1, “a” is present at upper of the figure. Please remove this.

Reviewer #3: The authors presented a nice piece of research on very interesting theme with sex-dependent effects. Here in the agriculture important insect, the mealybug Planococcus kraunhiae. The paper starts from a very interesting observation, a fascination in nature of the insects. The paper is well written and the research questions well formulated. I like that the discussion is balanced and critical. The research is done with high precision and quality.

I have two suggestions to the authors to improve their manuscript.

1. On the qPCR data, please write the number of biological and technical repeats done; although the work is done using a protocol as published before. Also the authors should confirm the stability in expression of the rpL32 as reference gene over the different stages tested. Why did the authors not sued 2 reference genes?

2. I appreciate that the authors did their upmost to measure the titer of ecdysteroids and even identify the different forms, however it did not work out successfully. It is not evident as I know from my own research. But the authors should also reflect in the discussion that the amount of free ecdysteroids can also depend on the process of conjugation, to bind or liberate ecdysteroid hormone and so reducing or increasing the titer concentration in the insect body.

6. PLOS authors have the option to publish the peer review history of their article (what does this mean?). If published, this will include your full peer review and any attached files.

Reviewer #1: No

Reviewer #2: No

Reviewer #3: No

---

## [Author Response · Author response to Decision Letter 0]

18 Mar 2020

We would like to thank the Editor and the Reviewers for reviewing our manuscript and for their comments. All the comments are very valuable and will improve our manuscript. Please find the responses to the comments below. In the revised manuscript, changes are highlighted with yellow.

Through careful proofreading, we noticed additional points that need to be revised, most of which are grammatical errors. These changes are highlighted with cyan.

Reviewer #1: 

Comments

Mealybugs are characterized by remarkable sexual dimorphic traits as a result of unusual diverging post-embryonic development. In some insect species, such sexual dimorphic traits were regulated by hormones such as JH and ecdysteroids. However, ecdysteroid titers have not been measured in the Japanese mealybug, Planococcus kraunhiae, and thus the involvement of ecdysteroids in sexual dimorphism in this species remains unknown.

In order to cultivate a better understanding for the role of ecdystderoids in sexual-dimorphic development in the mealybug, the authors first attempted to directly measure the ecdysteroid titers using LC-MS/MS. But they failed to detect ecdysteroids by their method. Therefore, the authors tried to estimate ecdysteroid titers by quantifying mRNA levels of the Halloween genes together with ecdysone response genes such as EcR and E75 by qRT-PCR. Finally, they propose that the changes in ecdysteroid titer are different between males and females, and that high ecdysteroid titer is essential for directing male adult development.

Overall, this is a nicely done, and several interesting observations are represented and discussed. But I think that there are several unignorable problems especially in their qRT-PCR data. 

Moreover, English used in this manuscript has a lot of problems. 

Response: Thank you for reviewing our paper. We carefully proofread the manuscript, and corrected the Result description.

Resolution of the figures is so poor that I cannot precisely recognize characters described on them. Consequently, this manuscript is not suitable for publication in this current form.

Response: Thank you for the comments. We checked the PDF file and found that the figures were in low resolution, but the high resolution original figures can be downloaded on the link at the top right of the figure pages.

Major requirements for revision

1. page 4, line 109. The authors said that they were not able to detect any ecdysteroid, probably due to the small size of the mealybug. If so, then the authors should increase the number of individuals subjected to the LC-MS/MS analysis.

Response: We agree with the Reviewer #1 that LC-MS/MS analysis of ecdysteroids with increased number of individuals will be desirable. However, in our trials to analyze ecdysteroids with LC-MS/MS, we used enough amount of pooled mealybugs (approximately 200 individuals, equivalent to ca. 10 mg) which was comparable to those used in other insect species such as the fruit fly Drosophila melanogaster. At present, it is unknown whether this is due to low titer of ecdysteroids in the mealybugs, or there is some unidentified humoral factors that inhibit the detection of ecdysteroids by LC-MS/MS. In any case, we believe that detection and quantification of ecdysteroids will be possible if the number of the individuals is dramatically increased. However, we are afraid that this is beyond the scope of the current manuscript, and will be performed in our future study.

2. page 9, lines 242-244 and Fig. S2. The authors identified an EcR ortholog of the mealybug, which showed highest similarity to EcR-A isoforms in other insect species. 

Were there any other EcR isoforms such as EcR-B1 and EcR-B2? 

If not, then the authors should show several lines of evidence that the mealybug examined in this study has only one isoform.

 And also, they should perform phylogenetic analysis as described in Fig. 1 to evaluate whether the PkEcR is indeed an ortholog of EcR in the mealybug.

Response: Thank you for the comment. We have only performed sequencing of a partial fragment of EcR and therefore did not identify all the isoforms. However, based on the alignment of the fragment that we sequenced (S2 Fig.), we are confident that the covered region matches EcR-A isoform.

EcR is a highly conserved gene with ligand-binding and DNA-binding domains (S2 Fig.). As opposed to the Halloween genes which are close paralogous copies and where a phylogenetic tree is interesting to perform to separate each of the copies, we do not think that providing a phylogenetic tree of the EcR sequence we identified is necessary.

3. Fig 2 and 3. How many trials did the authors perform qRT-PCR analyses? In general, researchers perform at least triplicate trials of qRT-PCRs and give SE to statistically assess the quantified data. The authors should provide such kind of information about statistical analysis on their qRT-PCR data.

Response: Thank you for the comment and we understand the concern. We used pooled individuals for each point, and did not do biological replicates for this experiment. In fact, we have another set of samples for analyzing developmental expression profiles, from the embryonic stage to the adults, that was used in our previous study (Vea et al., PLOS One 2016). For some of the genes including E93, we have confirmed that the profiles are consistent between two set of cDNA samples. However, it was not possible for us to combine the results from two sets of cDNA samples for calculating the average or SE because there was slight difference of the duration of each developmental stage: for example, the length of the the embryonic stage after oviposition and that of the first-instar nymphs are 10 days and 14–15 days respectively in the previous study, while they are 9–11 days and 11 days respectively in the present study. For these reasons, we showed the results from only one replicate in the manuscript.

4. page 10, lines 272-273. The authors said that expression levels of PkE75C and PkE75E were higher in males than females at the onset of male metamorphosis.

 But the expression level of PkE75C was specifically higher in females than males at day-4 embryo. 

The level was as same as that in male at pre-pupal stage. The authors should give some explanations about this female-specific increment of PkE75C expression level.

Response: Thank you for the comment. We agree with the Reviewer #1 that there is a female-specific increment of PkE75C at Day 4_Embryo. But we will not discuss on it in details because this study is focused on post-embryonic development. We revised the sentence to mention that there is a female-specific increment of PkE75C in the embryonic stage (L261–263 in the revised manuscript).

5. page 12, lines 343-345. The authors said that the sex-specific expression of PkE75 isoforms could be regulated by both ecdysteroids and JH. If so, then expression levels of PkE75 isoforms do not simply reflect ecdysteroids titers. This means that people cannot estimate ecdysteroid titers based on the expression levels of PkE75 isoforms. Therefore, the authors should directly quantify the ecdysteroid titers.

Response: As we pointed out in the Discussion, we used several markers to assess the titers of ecdysteroids, including PkE75. We agree that the potential interaction with other pathways may influence the transcriptional regulation of PkE75. We totally agree with the Reviewer #1 that direct quantification of ecdysteroids will be the best way to identify ecdysteroid titer. However, as we have stated above, this will be beyond beyond the current manuscript. In Discussion section (L370–380), we added a statement on quantifying ecdysteroid titers directly with increased number of individuals. 

Minor requirements for revision

1. Overall. All text describing experimental results should be written in the past tense. For example, in the line 9, the author described that "PkEcR expression is progressive and happens as small peaks at hatching". But they should change this phrase to "PkEcR expression was progressive and happened as small peaks at hatching.

Response: We corrected the description of the results into the past tense, as highlighted in yellow in the revised manuscript.

2. page 4, line 88. I think "a result of" should be better to change to "as a result of".

Response: Revised as suggested (L85).

3. page 4, line 97. Add period immediately after "[29]".

Response: Revised as suggested (L94).

4. page 4, line 101. I think "which is" should be better to insert prior to "involved in".

Response: Revised as suggested (L98).

5. page 4, line 107. "directs" should be replaced with "direct".

Response: Revised as suggested (L104).

6. page 6, line 166. "(" before [37] should be removed.

Response: Revised as suggested (L164).

Reviewer #2: Sexual dimorphism on metamorphosis is known in meal bug. Generally, juvenile hormone (JH) and ecdysteroid play crucial roles on regulation of polypenism. Authors has already showed the JH titer of Planococus kraunhiae. In this manuscript, authors estimate the ecdysteroid profiles of male and female in the meal bug, P. kraunhiae. It is impossible to measure the ecdysteroid titer directly because the bug is too small to get enough volume of hemolymph. The ecdysteroid titer was estimated from the expression profiles of 4 ecdysterodgenic genes and 2 ecdysone responsible genes. According to data, they discussed ecdysteroid titer in both sex and speculates the role of ecdysteorid on sexual dimophism. The manuscript is written well and statement is clear.

In general, I do not see serious problems on the scientific aspects of the manuscript. Thus, I’d add only some comments.

Response: Thank you for reviewing our paper.

1. In figure 5, ecdysteroid titer was estimated. The ecdysteroid level of female seems too low levels in N3 nymph. The expression levels of spo, dib and sad in N3 were similar to these in N2. What did you calculated the level is based on?

Response: We estimated the ecdysteroid titer in Figure 5 based on the expression profiles of Halloween genes and EcR, as described in L347–349.

2. Line 97. Followed by [97], please add full stop.

Response: A full stop was added (L94). Thank you for the comment.

3. Lin 100. What “the transcription factor” does indicate?

Response: We revised the sentence as “We further showed that the adult-specifying transcription factor E93” (L97–98), so that it is clear to the readers.

4. Line 107. I think it is mistype, please correct “the directi titers”.

Response: This was corrected as suggested (L104).

5. Line 166. Please remove “(“.

Response: This was corrected as suggested (L164).

6. Line 176. About what amount total RNA did you use for RT? Please mention.

Response: In our RNA isolation, we used glycogen as coprecipitant to increase the yield of RNA, as we described in our previous study (Vea et al., 2016). Since glycogen shows a UV absorbance and interfere RNA quantification with spectrophotometer, we were not able to quantify total RNA. Equal volume of total RNA were used for RT. 

7. In discussion section, from line 312 to 318 is very similar to line 319-330. Please rewrite these paragraphs.

Response: This part was rewritten and organized as suggested (L312–328).

8. In figure 1, “a” is present at upper of the figure. Please remove this.

Response: This was corrected as suggested. 

Reviewer #3: The authors presented a nice piece of research on very interesting theme with sex-dependent effects. Here in the agriculture important insect, the mealybug Planococcus kraunhiae. The paper starts from a very interesting observation, a fascination in nature of the insects. The paper is well written and the research questions well formulated. I like that the discussion is balanced and critical. The research is done with high precision and quality.

I have two suggestions to the authors to improve their manuscript.

Response: Thank you for reviewing our paper. We appreciate valuable comments.

1. On the qPCR data, please write the number of biological and technical repeats done; although the work is done using a protocol as published before. Also the authors should confirm the stability in expression of the rpL32 as reference gene over the different stages tested. Why did the authors not sued 2 reference genes?

Response: As we described in Response to Reviewer #1’s comments, we used pooled individuals for each point, and did not do biological replicates for this experiment. We have another set of samples for analyzing developmental expression profiles, from the embryonic stage to the adults, that was used in our previous study (Vea et al., PLOS One 2016). For some of the genes such as E93, we have confirmed that the profiles are consistent between two set of cDNA samples (data not shown in the manuscript).

Regarding adding another reference gene, we agree with the Reviewer #3 that using two reference genes would be desirable. This will enable us to quantify the transcript levels more precisely. In the present study we used rpL32 as the reference gene because this is one of the most frequently utilized one for qRT-PCR and had been routinely used for scale insects in particular. Since other reference genes have not been established in this species yet, if we consider adding a second reference gene, we would need to properly examine which candidate housekeeping gene would be the most appropriate as the reference gene: developmental expression profiles of many candidate genes would need to be examined and compared. Since it will take quite a long time to determine the second reference gene, we would rather not do it in the present study, but will consider this point in future studies.

2. I appreciate that the authors did their upmost to measure the titer of ecdysteroids and even identify the different forms, however it did not work out successfully. It is not evident as I know from my own research. But the authors should also reflect in the discussion that the amount of free ecdysteroids can also depend on the process of conjugation, to bind or liberate ecdysteroid hormone and so reducing or increasing the titer concentration in the insect body.

Response: Thank you for the suggestion. We added a paragraph regarding the possible conjugation of ecdysteroids in the hemolymph (L370–380) in Discussion section.

---

## [Editor Report · Decision Letter 1]

25 Mar 2020

Sex-specific expression profiles of ecdysteroid biosynthesis and ecdysone response genes in extreme sexual dimorphism of the mealybug Planococcus kraunhiae (Kuwana)

PONE-D-20-01381R1

Dear Dr. Minakuchi,

We are pleased to inform you that your manuscript has been judged scientifically suitable for publication and will be formally accepted for publication once it complies with all outstanding technical requirements.

With kind regards,

Christopher N. Boddy, Ph.D.

Academic Editor

PLOS ONE
---

## [Editor Report · Acceptance letter]

30 Mar 2020

PONE-D-20-01381R1 

Sex-specific expression profiles of ecdysteroid biosynthesis and ecdysone response genes in extreme sexual dimorphism of the mealybug *Planococcus kraunhiae* (Kuwana) 

Dear Dr. Minakuchi:

I am pleased to inform you that your manuscript has been deemed suitable for publication in PLOS ONE. Congratulations! Your manuscript is now with our production department. 

With kind regards,

on behalf of

Dr. Christopher N. Boddy 

Academic Editor

PLOS ONE